# Costunolide—A Bioactive Sesquiterpene Lactone with Diverse Therapeutic Potential

**DOI:** 10.3390/ijms20122926

**Published:** 2019-06-14

**Authors:** Dae Yong Kim, Bu Young Choi

**Affiliations:** 1Department of Biology Education, Seowon University, Cheongju, Chungbuk 361-742, Korea; kensei@hanmail.net; 2Department of Pharmaceutical Science & Engineering, Seowon University, Cheongju, Chungbuk 361-742, Korea

**Keywords:** costunolide, antioxidants, anti-inflammatory, anti-allergic, bone regenerating, neuroprotective, antimicrobial, hair growth promoting, anticancer, antidiabetic properties

## Abstract

Sesquiterpene lactones constitute a major class of bioactive natural products. One of the naturally occurring sesquiterpene lactones is costunolide, which has been extensively investigated for a wide range of biological activities. Multiple lines of preclinical studies have reported that the compound possesses antioxidative, anti-inflammatory, antiallergic, bone remodeling, neuroprotective, hair growth promoting, anticancer, and antidiabetic properties. Many of these bioactivities are supported by mechanistic details, such as the modulation of various intracellular signaling pathways involved in precipitating tissue inflammation, tumor growth and progression, bone loss, and neurodegeneration. The key molecular targets of costunolide include, but are not limited to, intracellular kinases, such as mitogen-activated protein kinases, Akt kinase, telomerase, cyclins and cyclin-dependent kinases, and redox-regulated transcription factors, such as nuclear factor-kappaB, signal transducer and activator of transcription, activator protein-1. The compound also diminished the production and/expression of proinflammatory mediators, such as cyclooxygenase-2, inducible nitric oxide synthase, nitric oxide, prostaglandins, and cytokines. This review provides an overview of the therapeutic potential of costunolide in the management of various diseases and their underlying mechanisms.

## 1. Introduction

Drug development from natural sources, particularly from plants, has long been the mainstay in medical management of various human ailments. A wide variety of non-nutritive plant constituents, commonly known as phytochemicals, are being used as therapy for many disease processes, including, but not limited to, infections, diabetes, heart diseases, neurological disorders, and cancer. In fact, it is estimated that about 40% of all medicines are natural compounds or their semisynthetic derivatives [1]. One of the major classes of bioactive phytochemicals is the terpenoids, which are widely present in various plants and marine organisms, and are being examined for developing new antifungal, anticancer, anti-inflammatory, and antiviral agents [2]. For example, artemisinin and paclitaxel are terpenoids used clinically as antimalarial and anticancer agents, respectively. The largest group of sesquiterpene lactones is germacranolides [3], which possesses a 10,5-ring structure and is present in several plant families. Germacranolides are key precursors of other sesquiterpene lactones with various polycyclic skeletons, such as guaianolides, eudesmanolides, etc. [4]. Costunolide, a colorless crystalline powder with a molecular formula of C_15_H_20_O_2_ and a molecular weight of 232.318 g/mol, is a well-known sesquiterpene lactone in the germacranolides series. This compound was first isolated from costus (*Saussurea lappa* Clarke) root and then isolated from various other plant species. [5]. Structurally, costunolide (Figure 1) is a monocarboxylic acid having three double bonds which by catalytic hydrogenation generates hexahydrocostunolide. Partial hydrogenation of costunolide produces dihydrocostunolide [6]. The bioactivity of costunolide is mediated through its functional moiety, α-methylene-γ-lactone, which can react with the cysteine sulfhydryl group of various proteins, thereby altering intracellular redox balance [5]. This review is aimed at summarizing the recent research on costunolide, focusing on its therapeutic potential, underlying mechanisms of action, and the prospect of using costunolide for future drug development.

## 2. Therapeutic Potential of Costunolide

### 2.1. Antioxidant and Anti-Inflammatory Effects of Costunolide

Oxidative stress resulting from cellular redox imbalance leads to many diseases, such as diabetes, atherosclerosis, and cardiovascular diseases [7]. The antioxidant activity of costunolide was studied in streptozotocin (STZ)-induced diabetic rat model, which demonstrated marked reduction in the levels of glutathione (GSH) in the brain, heart, liver, pancreas, and kidney. Oral administration of costunolide restored the GSH level in these tissues [8]. Increased levels of GSH may increase the levels of GSH-dependent enzymes, such as glutathione peroxidase (GPx) and glutathione-S-transferase (GST), thereby reducing tissue damage [9]. Oxidative stress oxidizes and damages membrane phospholipid to produce lipid peroxides, such as malondialdehyde (MDA) and hydroxynonenals (HNE), which by forming DNA adducts may cause oxidative tissue damage. Costunolide also decreased lipid peroxidation levels and increased in SOD, catalase, and GPx activity in MCF-7 & MDA-MB-231 cells [10]. In a rat intestinal mucositis (IM) model, administration of costunolide restored 5-floirouracil (5FU)-depleted plasma superoxide dismutase (SOD) levels in rat intestinal mucosa [11]. Costunolide also abrogated hydrogen peroxide (H_2_O_2_)-induced ROS production in rat pheochromocytoma (PC12) cells [12].

Persistent tissue inflammation plays an important role in the pathogenesis of various infectious and noninfectious diseases, such as rheumatoid arthritis, Alzheimer’s disease, and arteriosclerosis [13]. Costunolide exhibited anti-inflammatory properties in a number of preclinical studies. The compound attenuated carrageenan-induced paw edema, myeloperoxidase (MPO) activity and *N*-acetylglucosaminidase (NAG) activity in mice [13]. One of the transcriptional regulators of proinflammatory gene expression is the transcription factor nuclear factor-kappaB (NF-κB). Costunolide negated NF-κB activation via blockade of IκBα phosphorylation in lipopolysaccharide (LPS)-stimulated RAW264.7 cells, thereby reducing the expression of proinflammatory markers, such as inducible nitric oxide synthase (iNOS), and the production of nitric oxide (NO) [14]. Chen et al. also demonstrated that treatment with costunolide inhibited 5-fluorouracil (5-FU)-induced expression of iNOS, cyclooxygenase-2 (COX-2), TNF-α, and the production of nitric oxide (NO) in a mouse model of intestinal mucositis by blocking the activation of NF-κB [11]. Costunolide diminished STAT1 and STAT3 phosphorylation in IL-22 or IFN-γ-induced human keratinocytes [15]. Likewise, treatment of human THP-1 cells with costunolide inhibited interleukin (IL)-6-induced phosphorylation and the DNA binding activity of signal transducer and activator of transcription (STAT)-3 via downregulation of Janus-activated kinase (JAK)-1 and -2 [16]. Moreover, costunolide showed an anti-inflammatory effect as evidenced by amelioration of ethanol-induced gastric ulcers in mice. This study also reported that the compound suppressed the activation and/or induction of NF-κB, TNF-α, NO, iNOS, and COX-2 [17]. Costunolide inhibited interleukin (IL)-1β protein and mRNA expression in LPS-stimulated RAW264.7 cells by blocking activator protein (AP-1) transcriptional activity via downregulation of mitogen-activated protein kinase (MAPK) phosphorylation [18]. In addition, costunolide alleviated lung inflammation in carrageenan-induced mouse pleurisy model as evidenced by reduced accumulation of polymorphonuclear cells and reduced expression of TNF-α, intracellular adhesion molecule-1 (ICAM-1), P-selectin, and nitrotyrosine [19].

Heme oxygenase-1 (HO-1) has been reported to mediate anti-inflammatory and cytoprotective activity [20]. Pae and colleagues [21] have reported that the production of TNF-α and IL-6 in LPS-stimulated RAW264.7 cells was decreased by treatment with costunolide, which increased the expression and activity of HO-1 via enhanced nuclear accumulation of a redox-regulated transcription factor, nuclear factor erythroidrelated factor-2 (Nrf2). Pretreatment with a HO-1 inhibitor abrogated the inhibitory effect of costunolide on LPS-induced TNF-α and IL-6 production [21]. CD4^+^ T cell activation and proper differentiation into T helper (Th) cells are important for establishing an adaptive immune response against foreign pathogens. However, an excessive activation of Th cells leads to inflammation and autoimmune diseases [22]. When CD4^+^ T cells were induced to differentiate, costunolide markedly reduced the differentiation into a population of Th1, Th2, and Th17 subsets. Costunolide also inhibited the expression level of Th subset-polarizing master genes such as T-bet, GATA3, and RORγt. The compound reduced the level of CD4^+^ T cell activation marker CD69 and attenuated T cell proliferation by blocking phosphorylation of extracellular signal-regulated kinase (ERK) and p38 MAPK [23]. 

### 2.2. Anti-Allergic Effects of Costunolide

Chemokines play an important role in inducing various allergic and inflammatory skin diseases, such as atopic dermatitis, psoriasis, and eczema. Keratinocytes are known to respond to various chemokines, such as chemokine (C-C motif) ligand (CCL)-17 (also known as TARC), CCL-22 (alternatively known as MDC), CCL-5 (synonym RANTES), and IL-8, which are involved in precipitating atopic dermatitis [24]. Costunolide significantly reduced mRNA expression of various chemokines including TARC/CCL17, MDC/CCL22, RANTES/CCL5, and IL-8 in HaCaT cells stimulated with TNF-α and IFN-γ [25]. In the OVA-induced asthmatic mouse model, costunolide reduced eosinophil infiltration, inflammation score, and mucin secretion in the lungs. In particular, the increase of eosinophil count in BALF (bronchoalveolar lavage fluid) by OVA was significantly inhibited by costunolide. Moreover, the compound decreased the expression and secretion of Th2 cytokines (IL-4 and IL-13) in BALF and lung tissue [26]. Costunolide reduced the activity of β-hexosaminidase, an enzyme involved in mast cell degranulation, and decreased IL-4 mRNA transcript in IgE-sensitized rat basophilic leukemia (RBL-2H3) cells. In addition, the inhibition of IL-5-dependent growth of Y16 pro-B cells suggests the potential of costunolide or its derivatives to be developed as mast cell stabilizers and pro-B cell proliferation inhibitors in allergic diseases [26,27].

### 2.3. Costunolide in Bone Remodeling

Osteoporosis, a disease of the bone generally characterized by excessive bone resorption due to poor osteoblastic and enhanced osteoclastic activity, is very common among the elderly population. Although few therapeutic interventions, such as the use of calcium and vitamin D3, parathyroid hormone analogs, bisphosphonates, and monoclonal antibodies, are current clinical recommendations, there is emerging need of developing new drugs [28]. Several studies have demonstrated the potential of costunolide in improving bone health. Costunolide stimulated the growth and differentiation of murine osteoblastic cells (MC3T3-E1) cells, as characterized by increased alkaline phosphatase (ALP) activity, collagen deposition and mineralization. These osteoblastic activity of costunolide was abrogated by cotreatment with pharmacological inhibitors of either estrogen receptor (ER) or phoaphatidylinositol-3-kinase (PI3K), suggesting that the increased mineralization by the compound was associated with increased activation of ER and PI3K [29]. Likewise, costunolide increased the ALP activity and matrix mineralization, and elevated the transcription of a number of differentiation factors, such as distal-less homeobox 5 (Dlx5), runt-related transcription factor 2 (Runx2), and osteocalcin (OC) in mouse mesenchymal stem cell (C3H10T1/2) by stimulating activated transcription factor 4 (ATF4)-dependent increased expression and activity of HO-1. The blockade of HO-1 by treating cells with tin (IV) protoporphyrin IX dichloride (SnPP) blocked costunolide-induced Runx2 expression, suggesting that costunolide-induced osteoblast differentiation is regulated by ATF4-dependent HO-1 expression [30]. The receptor activator of nuclear factor kappa-B ligand (RANKL) induces the differentiation of bone marrow-derived macrophages into osteoclasts, a key mediator of bone resorption. Treatment with costunolide inhibited osteoclast differentiation by blocking the expression of nuclear factor of activated T cells, cystoplasic-1 (NFATc1) through the inhibition of c-Fos transcriptional activity, without affecting c-Fos expression. The compound also attenuated the mRNA expression of tartrate-resistant acid phosphatase (TRAP) and osteoclast-associated receptor (OSCAR) (Figure 2). Thus, costunolide inhibited RANKL-induced osteoclast differentiation by inhibiting c-Fos transcriptional activity [31].

### 2.4. Costunolide as a Neuroprotective Agent

Parkinson disease (PD) is one of the neurodegenerative diseases characterized by reduced dopaminergic (DAergic) neuronal transmission in the substantia nigra (SN). One of the key regulators of DAergic nerve transmission, especially the regeneration of synaptic vesicles, and the storage, metabolism, and release of DA at nerve endings is α-synuclein (ASYN), which is transcriptionally regulated by nuclear receptor related-1 (Nurr1). In PD patients with Nurr1 mutations, Nurr1 expression was decreased and ASYN expression was increased. Whereas the expression of the Nurr1 gene is essential for the development and maintenance of nigral DAergic neurons, overexpression of ASYN causes selective degeneration and toxicity of DAergic neurons. Nurr1 also participates in DA metabolism by regulating vesicular monoamine transporter type 2 (VMAT2) and dopamine transporter (DAT) [32]. Ham et al demonstrated that costunolide inhibited DA-induced apoptosis of human neuroblastoma (SH-SY5Y) cells which was associated with decreased ASYN expression and the restoration of DA-mediated reduced Nurr1, VMAT2, and DAT level [33]. These results suggest the potential of costunolide in the management of PD. Since oxidative stress-mediated neuronal cell death is a well-known cause of many neurodegenerative diseases, the reduction of reactive oxygen species (ROS) can be a pragmatic approach to delay the disease progression. Treatment with costunolide inhibited H_2_O_2_-induced apoptosis of PC12 cells by reducing intracellular ROS, stabilizing mitochondrial membrane potential (MMP), and decreasing the caspase-3 activity (Figure 3). Moreover, costunolide reduced H_2_O_2_-induced cell death by blocking the phosphorylation of p38 MAPK and ERK [12]. Besides oxidative stress, persistent inflammation often leads to neurodegeneration. By virtue of its anti-inflammatory properties costunolide inhibited LPS-induced apoptosis of BV2 microglial cells by decreasing the expression of a series of neuroinflammatory mediators, such as TNF-α, IL-1, IL-6, iNOS, macrophage chemoattractant protein-1 (MCP-1), and COX-2 via the inhibition of NF-κB and MAPK activation [34].

### 2.5. Antimicrobial Properties of Costunolide

Several studies have demonstrated the antimicrobial activity of costunolide (Table 1). The antibacterial activity of costunolide against *Mycobacterium tuberculosis* H37Rv (*M. tuberculosis*) [35] and *Mycobacterium avium* (*M. avium*) [36] in fluorometric Alamar Blue microassay and radiorespirometric bioassay, respectively, suggest that the compound may be considered for developing antitubercular drugs. In addition, in vitro agar diffusion test showed that costunolide exhibited antimicrobial activity against *Staphylococcus aureus* (*S. aureus*), *Escherichia coli* (*E. coli*), and *Pseudomonas aeruginosa* (*P. aeruginosa*) [37]. Costunolide also inhibited the growth of *H. pylori* [38], which is causally linked with gastric and duodenal ulcers. In vitro disc diffusion assay revealed that costunolide inhibited the growth of various pathogenic fungi, such as *Trichophyton mentagrophytes*, *T. simum*, *T. rubrum*, *Epidermophyton floccosum*, *Scopulariopsis* sp., *Aspergillus niger*, *Curvulari lunata*, *Magnaporthe grisea*, and *Candida albicans* [39]. Costunolide also showed antifungal activity against *Botauttis cinereal*, *Colletotrichum acutatum*, *Colletotrichum fragariae* and *Colletotrichum gloeosporioides* [40], and *C. echinulata* [41]. The antiviral property of costunolide was evident from the inhibition of hepatitis B surface antigen (HBsAg) expression in human hepatoma Hep3B cells and that of hepatitis B e antigen (HBeAg), a hepatitis B virus genome replication marker, in human hepatocytes and HepA2 cells [42]. 

Lipoteichoic acid (LTA)-induced acute lung injury (ALI) in mice is a model to represent experimental pneumonia. Treatment with costunolide significantly reduced LTA-induced inflammatory cell infiltration and lung tissue damage by decreasing the production of various cytokines and chemokines. Moreover, the compound inhibited LTA-induced iNOS expression in mouse bone marrow-derived macrophages by blocking phosphorylation of TAK1, p38 MAPK, and ERK, without affecting the activation of NF-κB [43]. Thus, costunolide may be considered as a lead compound for developing novel antimicrobial agents.

### 2.6. Costunolide in the Treatment of Alopecia

The cosmetic use of herbal products, especially for preventing hair loss or promoting hair growth, have long been practiced throughout the world. The herbal therapies used as hair growth promoters are expected to have low toxicity, be easy to use, low cost, and have high patient compliance. As the physiological and biochemical pathways in hair follicle dermal papillary cells (hHFDPCs) are unfolded, the mechanistic basis of hair growth promotion by many natural products is being explored [44,45]. It has been reported that topical application of costunolide significantly improved hair growth in C57BL/6 mice in vivo and promoted the proliferation of hHFDPCs in vitro [46]. Mechanistically, costunolide inhibited 5α-reductase activity and suppressed transforming growth factor (TGF-β1) induced phosphorylation of Smad-1/5 (mothers against decapentaplegic-1/5) in hHFDPCs, whereas the compound increased the levels of β-catenin and Gli1 mRNA and protein [46]. Thus, the development of costunolide-based formulation for the treatment of alopecia would be an interesting approach pending further studies on the toxicity and pharmacokinetic properties of the compound.

### 2.7. Costunolide as an Anticancer Agent

The search for anticancer agents from natural sources, especially plants, has led to the discovery of many clinically useful drugs. Extensive investigation of the anticancer effects of costunolide have shown that the compound induces apoptosis and inhibits proliferation of various cancer cells in vitro, and suppresses angiogenesis and metastasis. The following section will shed light on the biochemical processes and molecular targets of costunolide in exerting its anticancer effects. 

#### 2.7.1. Inhibition of Cell Proliferation

Costunolide decreased the proliferation of various cancer cells including those of the colon, breast, prostate, liver, gastric, and blood cancer cells [47,48,49,50,51]. Treatment of HCT-116 cells with costunolide decreased cell proliferation by inhibiting phosphorylation of mammalian target of rapamycin (mTOR) and its downstream kinases p70S6K and 4E-BP1, and increasing the phosphorylation and nuclear localization of p53 [52]. The antiproliferative effect of costunolide was mediated, at least in part, through suppression of cellular glutaminolysis via blockade of the promoter activity of glutaminase 1 (GLS1). The compound also decreased GLS1 mRNA and protein expression in p53-dependent manner since pretreatment with a p53 inhibitor reversed costunolide-mediated suppression of GLS1 activity and expression [52]. The antiproliferative effect of costunolide in MCF-7 cells was associated with microtubule polymerization and alteration of spindle morphology. [47].

Several other studies have reported that costunolide inhibits various tumor cell proliferation by blocking G2/M phase of the cell cycle and modulating the effect of cyclins and cyclin-dependent kinases (Cdk) [51,53,54]. The growth inhibition of SW-480 cells upon treatment with costunolide was associated with the downregulation of cyclin D1 and survivin, which was mediated via inhibition of nuclear translocation of β-catenin and its co-activator molecule galectin-3 [49]. Peng et al. [55] reported that costunolide induced cell cycle arrest at G2/M phase in MCF-7 and MDA-MB-231 cells via activation of p53 and p-14-3-3 expression and inhibition of c-Myc, p-AKT, and p-BID expression. Moreover, the ratio of BAX/BCL-2 was significantly increased upon costunolide treatment, which led to the induction of apoptosis in these cells [55]. Another study showed that costunolide-induced G2/M cell cycle arrest in MDA-MB-231 cells, which was mediated through the inhibition of Cdc2 and cyclin B1, and the elevation of p21^WAF1^ expression was independent of p53 activation [56]. Roy and colleagues have demonstrated that the G2/M phase of cell cycle arrest in MCF-7 and MDA-MB-231 cells incubated with costunolide was mediated through the downregulation of cell cycle regulatory proteins, such as cyclin D1, D3, CDK-4, CDK-6, p18^INK4c^, p21^CIP1/Waf-1^, and p27^KIP1^. However, the compound did not affect the proliferation of normal mammary epithelial (MCF-10A) cells [57]. Likewise, the increase level of p21^WAF1^ and reduced expression of cyclin B1 and CDK2 by costunolide led to the G2/M phase arrest in K562 cells. According to this study, the compound enhanced imatinib-induced apoptosis in K562 cells via modulation of B cell receptor (Bcr)/Abl and STAT5 signaling pathways. In another study, these authors reported that costunolide sensitized K562 cells to doxorubicin via inhibition of the PI3K/Akt activity [58]. In another study, costunolide arrested the cell cycle at the G2/M phase through the downregulation of Chk2/Cdc25c/Cdk1/cyclin B1 signaling in human hepatoma HA22T and VGH cells [51]. Incubation of human prostate cancer (PC-3, DU-145, and LNCaP) cells with costunolide arrested the cell cycle at the G1 phase, which was associated with the inhibition of the CDK2 activity and Rb phosphorylation [50]. Moreover, costunolide upregulated p53 and p21 expression in human esophageal squamous Eca-109 cells, thereby inducing G1/S phase arrest [59]. 

#### 2.7.2. Induction of Apoptosis

##### Mitochondria-Mediated Apoptosis 

Costunolide induced mitochondria-mediated apoptosis as evidenced by the inhibition of Bcl-2, induction of Bax, and release of cytochrome c in human prostate (PC3 and DU-145) [60], leukemia (K562) [48], oral cancer (Eca-109) [59], gastric cancer (SGC-7901) [54], lung squamous carcinoma (SK-MES1) [53], and bladder cancer (T24) [61] cells. Treatment of PC3 and DU-145 cells with costunolide led to the generation of ROS, the phosphorylation of c-Jun-N terminal kinase (JNK) and p38 MAPK, the inhibition of Bcl-2 and Bcl-xl, and the induction of Bax, thereby leading to reduced mitochondrial membrane potential and cytochrome c release and caspase 3 activation. The apoptosis induction by costunolide resulted in the reduced growth of PC3 cells xenograft tumors in nude mice [60]. Likewise, costunolide induced mitochondria-mediated apoptosis by upregulation of Bax, downregulation of Bcl-2, and activation of caspase-3 and poly ADP-ribose polymerase via ROS production and loss of mitochondrial membrane potential in oral cancer Eca-109 cells [59]. The compound also activated JNK in human leukemic U937 cells, thereby leading to mitochondrial cell death via phosphorylation of Bcl-2 and translocation of Bax to mitochondria [62]. Hua et al. demonstrated that the apoptosis of SK-MES-1 cells upon treatment with costunolide was mediated through upregulation of p53 and Bax expression, downregulation of Bcl-2 expression, and caspase-3 activation [53]. In addition to inducing Bax, caspase-3 and PARP cleavage in T24 bladder cancer cells, costunolide also attenuated expression of survivin and Bcl2 as a mechanism of apoptosis induction [61]. Costunolide induced mitochondrial-mediated apoptosis through activation of caspase-3, -8, and -9 in ovarian cancer cell lines (MPSC1PT, A2780PT, and SKOV3PT) and the human endometriotic epithelial cells (11Z, 12Z) [63,64]. Furthermore, incubation of multidrug resistant ovarian cancer cells (OAW42-A) with costunolide attenuated cell growth with an IC_50_ of 25 µM, and induced apoptosis, which was mediated through induction of Bax, decreased the expression of Bcl-2 and cleavage of caspase-3 and 9. Moreover, the compound induced autophagy as evidenced by the elevated expression of LC3 II and Beclin 1 [65]. 

##### Endoplasmic Reticulum (ER) Stress-Mediated Apoptosis

Apoptosis may result from continued ER stress that activates unfolded protein response (UPR) signaling pathways. Costunolide activated the ionositol requiring enzyme (IRE)-1α, a resident ER membrane protein, which further activated JNK by recruiting adapter molecules TRAF2 and ASK1 in cultured lung adenocarcinoma cell line A549 cells [66]. Costunolide-activated JNK led to Bcl-2 phosphorylation at serine 70, a mechanism to convert antiapoptotic Bcl-2 to play proapoptotic functions, thereby causing cytochrome c release, caspase-3 activation, and PARP cleavage, leading to induction of apoptosis. Authors have further demonstrated that costunolide-induced ROS generation played a critical role in this process as pretreatment of cells with ROS scavenger *N*-acetyl cysteine abrogated costunolide-induced ER stress and apoptosis [66]. A similar mechanism of ROS-mediated ER stress induction by costunolide led to the expression of Bip and IREα, and the activation of the JNK pathway, leading to apoptosis in human osteosarcoma U2OS cells [67]. Recent studies have shown that the thioredoxin/thioredoxin reductase (TrxR) system causes tumor cell resistance to oxidative stress-induced apoptosis. Surface plasmon resonance analysis and molecular docking study revealed that costunolide directly interacted with TrxR1 via its lactone oxygen atom with Gln-494 of TrxR1 and inhibited the activity of TrxR1, thereby increasing the production of ROS and inducing ROS-dependent ER stress and apoptosis in colon cancer cells (HCT-116, SW-620, and HT-29 cells). This study also demonstrated that the compound arrested G2/M phase of cell cycle and attenuated the expression of cyclin B1, CDC2, MDM2, and Bcl2 and increased the expression of Bax and cleavage of caspase 3, which was reversed by cotreatment with *N*-acetyl cysteine, suggesting the involvement of ROS in costunolide-induced retardation of tumor cell growth. Furthermore, costunolide treatment of mice transplanted with colon cancer cells inhibited tumor growth and decreased TrxR1 activity and ROS levels [68].

##### Death Receptor-Mediated Apoptosis

Extrinsic mechanisms of apoptosis induction by costunolide have also been reported. The induction of apoptosis in estrogen receptor-negative human breast cancer (MDA-MB-231) cells by costunolide involves the activation of Fas, caspase-8, caspase-3, and the degradation of PARP [56]. Costunolide also increased the phosphorylation of Fas-associated death domain (FADD) at serine 194, leading to apoptotic cell death in human B cell lymphoma. [69].

#### 2.7.3. Telomerase Reverse Transcriptase (TERT) Inhibition

Telomeres, which maintain genomic integrity in normal cells, are shortened upon each cell division, thus leading to chromosomal instability, cellular senescence, and aging. However, the length of the telomeres is maintained by high levels of telomerase enzyme present in tumor cells, thereby allowing cancer cells to be immortal. Therefore, telomerase has been considered as a possible target for cancer treatment [69]. It has been reported that costunolide caused significant inhibition of telomerase activity in human B cell leukemia (NALM-6) cells by decreasing the mRNA and protein expression of human telomerase reverse transcriptase (hTERT), which controls the enzymatic activity of telomerase, and induced apoptosis in these cells [68]. Likewise, costunolide showed strong inhibition of telomerase activity in MCF-7 and MDM-23-231 cells through the downregulation of hTERT mRNA via inactivation of c-Myc and Sp1 transcription factors [69]. The antiproliferative and apoptosis-inducing effects of costunolide was also associated with inhibition of hTERT in human glioma cells [70,71] and human hepatocellular carcinoma (HepG2/C3A, PLC/PRF/5) (A172, U87MG) cells [72].

#### 2.7.4. Inhibition of Angiogenesis

The persistent growth and spread of a tumor require a constant supply of nutrients and oxygen to the cancer cells. The formation of new blood vessels, a process known as angiogenesis, is therefore an essential step in tumor invasion and metastasis. The discovery of angiogenesis inhibitors (e.g., avastin) helps reduce the morbidity and mortality from various cancers. A key angiogenic molecule is vascular endothelial growth factor (VEGF), which by binding with VEGF receptors (VEGFR) on vascular endothelial cells promotes formation of new blood vessels [73].

In a murine cannulated sponge implant angiogenesis model, administration of costunolide to Swiss albino mice implanted with polyester polyurethane sponges used as a framework for fibroblast tissue growth reduced the levels of VEGF and hemoglobin content in fibrovascular tissue, suggesting the antiangiogenic property of the compound [74]. Jeong et al. reported that costunolide attenuated VEGF-induced proliferation and chemotaxis of human umbilical vein endothelial cells (HUVECs), and blocked VEGF-induced phosphorylation of KDR/Flk-1 in NIH 3T3 cells overexpressing KDR/Flk-1. In addition, VEGF-stimulated neovascularization in mouse corneal micropocket analysis was reduced by costunolide treatment [75]. In another study, costunolide significantly reduced VEGF secretion and decreased VEGF mRNA levels in human gastric cancer (AGS), colon cancer (Caco-2), and liver cancer (HepG2/C3A) cells. This study also reported that costunolide significantly reduced VEGFR1 and VEGFR2 expression at both mRNA and protein levels [76]. 

#### 2.7.5. Inhibition of Tumor Metastasis

Cancer metastasis refers to the spread of tumor cells from their site of origin to other distant parts of the body. Metastasis consists of multistep processes including tumor cell spread, extracellular matrix (ECM) degradation, tumor cell invasion in ECM, angiogenesis, and secondary metastatic tumor growth [77]. Costunolide inhibited TNF-α-induced migration and invasion of MDA-MB-231 breast cancer cells by downregulating the expression of matrix metalloproteinase (MMP)-9 gene via blockade of NF-κB activation. Moreover, the xenograft tumor growth of MBA-MB-231 cells in athymic nude mice was diminished upon treatment with costunolide [77]. The MMP-2 and MMP-9 are key molecules involved in tumor invasion and metastasis. Costunolide significantly inhibited invasion and decreased MMP-2 expression in human neuroblastoma (NB-39) cells [78]. The invasion of soft tissue sarcoma (TE-671, SW-872, and SW982) cells was also inhibited by costunolide via modulation of MMPs expression [79]. Of the various forms of metastasis, lymphatic metastasis is an important determinant in cancer therapy and staging. Costunolide inhibited the proliferation and capillary formation of TR-LE (temperature-sensitive mouse lymphoid endothelial cells) cells, suggesting that the compound can provide clinical benefits as an inhibitor of lymphoproliferative growth during tumor metastasis [80]. Epithelial–mesenchymal transition (EMT) is critical step in tumor invasion and metastasis. One of the mechanisms that the EMT process initiates in tumor invasion is the detyrosination of tubulin via inhibition of tyrosine ligase, and the detyrosinated tubulin forms microtentacles (McTN) which promotes tumor cell reattachment to the endothelial layer during tumor invasion. Costunolide significantly reduced detyrosinated tubulin and the frequency of McTN in multiple invasive breast tumors, thereby preventing tumor cells attachment with endothelial tissue and blocking invasion [81].

### 2.8. Antidiabetic Effects of Costunolide

An in vitro assay has revealed that the methanol extract of leaves of *Costus speciosus* inhibited α-glucosidase activity with an IC_50_ value of 67.5 μg/ml and attenuated α-amylase activity with an IC_50_ value of 5.88 mg/ml, which is lower than the reference compound acarbose [82]. Since costunolide is abundantly present in leaves of *Costus speciosus*, this study indicates the potential of costunolide in managing glycemic control. A subsequent study demonstrated that costunolide significantly reduced blood glucose level, glycosylated hemoglobin (HbA1c), serum total cholesterol, triglyceride, and LDL cholesterol level in streptozotocin (STZ)-induced diabetic rats [83]. Moreover, the compound remarkably increased plasma insulin, tissue glycogen, HDL cholesterol, and serum protein level [83]. Since oxidative stress affect the pathogenesis and progression of diabetic tissue injury, the induction of antioxidant enzymes, such as glutathione peroxidase, catalase, and superoxide dismutase in STZ-induced diabetic rat’s pancreas indicates the role of costunolide in improving glycemic control in diabetes [8]. However, additional studies are warranted to ascertain the antidiabetic property of this compound. 

## 3. Pharmacokinetics and Toxicity Profile

Pharmacokinetic studies are an integral part of the drug discovery process. The understanding of the absorption, distribution, metabolism, and elimination of the drug-to-be is an essential step in new drug development. There have been several studies reporting the pharmacokinetic profile of costunolide. The maximum plasma concentration (C_max_) and time required to attain highest plasma level of the molecule (T_max_) after oral administration of costunolide to Wistar rats were found as 0.024 ± 0.004 mg/L and 9.0 ± 1.5 h, respectively. The half-life (t_1/2_) and area under the curve (AUC) were 4.97 h and 0.33 ± 0.03 mg·h/mL, respectively [84]. However, a subsequent study reported that after oral administration of costunolide to Wistar rats, the C_max_ and T_max_ were 19.84 ng/mL and 10.46 h, respectively, and the half-life (t_1/2_) and AUC were 5.54 h and 308.83 ng·h/mL, respectively [85]. According to a recent study, oral administration of costunolide to SD rats showed C_max_, T_max_, t_1/2_, and AUC as 0.106 ± 0.045 μg/mL, 8.00 h, 14.62 ± 3.21 h, and 1.23 ± 0.84 μg·h/mL, respectively [86]. The large variation in pharmacokinetic parameters between these studies may be due to the use of different assay techniques and/or animal models. In addition, intravenous administration of costunolide to Sprague–Dawley rats revealed the C_max_ as 12.28 ± 1.47 μg/mL, and the half-life (t_1/2_) and AUC were detected as 1.16 ± 0.06 h and 3.11 ± 0.13 μg·h/mL, respectively [87]. These results would have immense importance in further development of costunolide-based therapy. 

Although costunolide has been examined extensively for its therapeutic potential in various animal models as discussed in the previous sections of this review, acute and chronic toxicity studies are scarce. A recent study demonstrated that the compound induced apoptosis in normal Chinese hamster ovarian cells by inducing clastogenic and genotoxic effects as evidenced by micronuclei formation and chromosomal breaks [88]. Thus, more rigorous toxicity studies to determine the lethal dose (LD_50_) and ensure safety of the compounds is of paramount importance in further progressing the development of costunolide as a drug candidate. 

## 4. Conclusions

Sesquiterpene lactones form a large, structurally diverse group of natural products found almost universally in plants. Extensive investigation of the therapeutic potential of sesquiterpene lactones has yielded important candidates for pharmaceutical development [89]. Costunolide is a well-known sesquiterpene lactone, which has been isolated from various plant species. As has been discussed in earlier sections, costunolide has been reported to possess antioxidant, anti-inflammatory, antiallergic, bone remodeling, neuroprotective, antimicrobial, hair growth promoting, anticancer, and antidiabetic properties (Figure 4 and Table 2). Limited pharmacokinetic studies have also shown that the compound can be bioavailable. However, the majority of these studies have been conducted in cultured cells or using an in vitro system. Considering the therapeutic value of the compound, it would be interesting to further examine the effects of costunolide in various other animal models to reveal the subacute and chronic toxicities, detailed elucidation of molecular mechanisms of action, and structural modifications to develop new therapeutics based on costunolide or its derivatives.

## Figures and Tables

**Figure 1 ijms-20-02926-f001:**
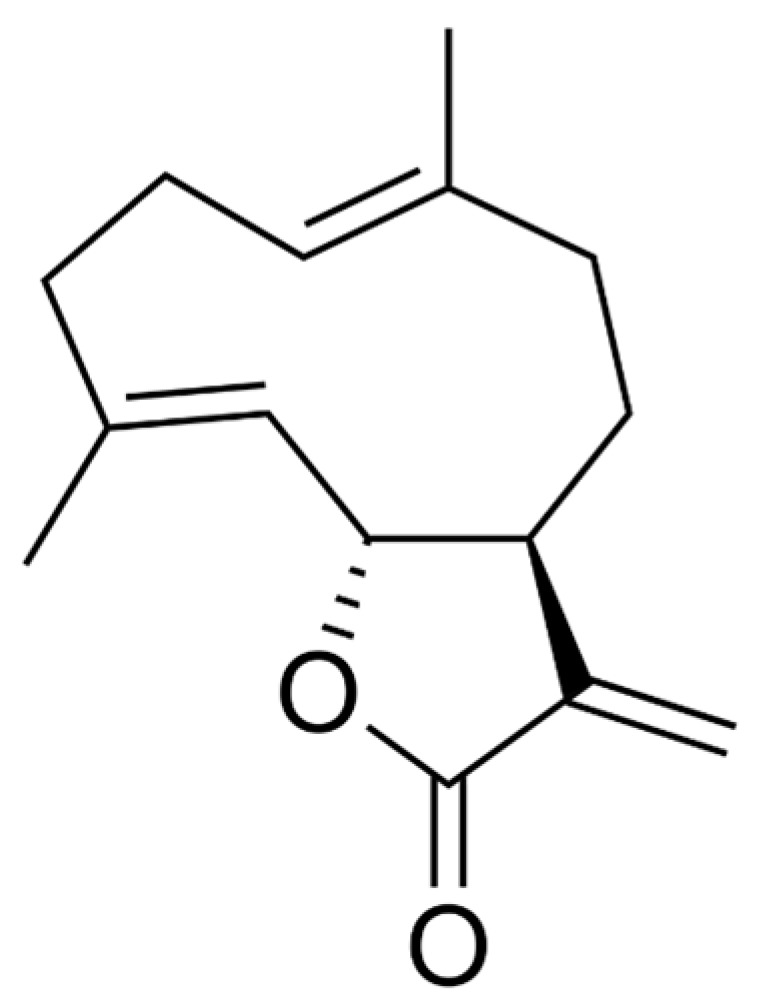
Chemical structure of costunolide.

**Figure 2 ijms-20-02926-f002:**
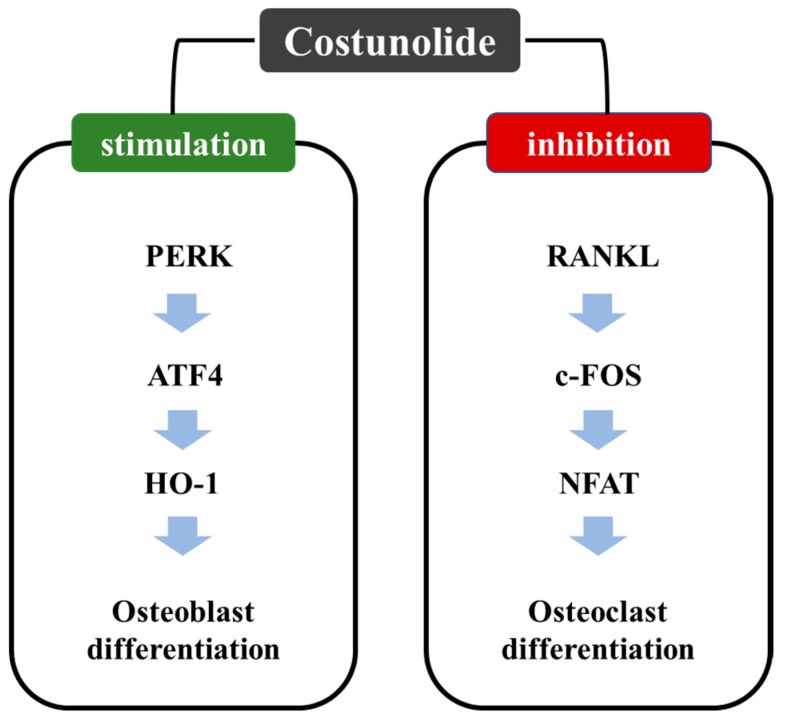
Effect of costunolide on differentiation of osteoblast and osteoclast. Costunolide induces osteoblast differentiation through ATF-4-induced HO-1 expression in mesenchymal stem cells. On the other hand, costunolide suppressed RANKL-mediated osteoclast differentiation via inhibiting RANKL-mediated c-Fos transcriptional activity in bone marrow cells.

**Figure 3 ijms-20-02926-f003:**
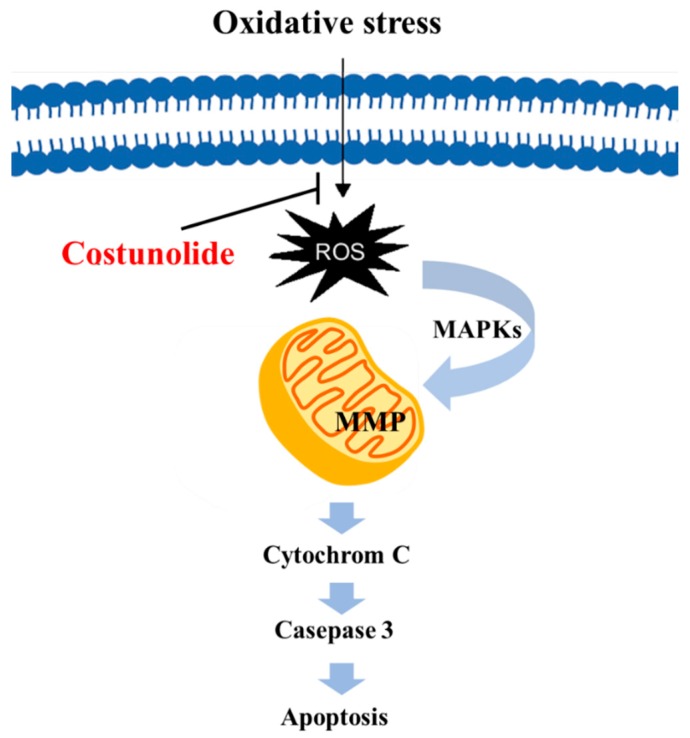
The effect of costunolide on apoptosis of neurons. Costunolide reduced intracellular ROS caused by oxidative stress. As a result, mitochondrial membrane potential (MMP) stabilized and apoptosis-related proteins such as caspase 3 decreased.

**Figure 4 ijms-20-02926-f004:**
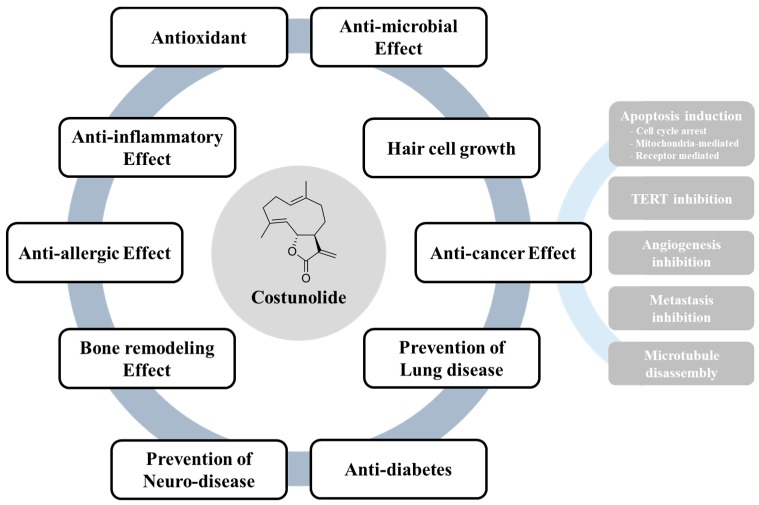
Bioactivities of costunolide. Costunolide could exert its therapeutic potential including antioxidant, anti-inflammatory effect, anti-allergic effect, bone remodeling effect, prevention of neurodegenerative disease, anti-microbial effect, inhibition of alopecia, prevention of lung disease and anti-diabetic effect. In particular, costunolide elicits anti-cancer activities partly through induction of apoptosis, Inhibition of cell proliferation, TERT, angiogenesis, metastasis and microtubule disassembly.

**Table 1 ijms-20-02926-t001:** Antimicrobial activity of costunolide.

Effect	Tested Organisms		Concentration	Reference
Antibacterial activity	*M. tuberculosis*	MIC (mg/L)	12.5	[35]
	*S. aureus* *E. coli* *P. aeruginosa*	MIDZ (mm)	18 19 14	[37]
	*M. avium* *M. tuberculosis*	MIC (μg/mL)	128 32	[36]
	*H. pylori*	MIC (μg/mL)	100–200	[38]
Antifungal activity	*Trichophyton mentagrophytes* *T. simum* *T. rubrum* *Epidermophyton floccosum* *Scopulariopsis sp.* *Aspergillus niger* *Curvulari lunata* *Magnaporthe grisea*	MIC (µg/mL)	62.5 62 31 or 62 125 250 125 250	[39]
	*Colletotrichum acutatum* *Colletotrichum fragariae*	MDIZ (mm)	4 6	[40]
	*C. echinulata*	EC_50_ (μg/mL)	6	[41]
Antiviral activity	*Hepatitis B virus (HBV)*	IC_50_ (μM)	1	[42]

MIC: minimum inhibitory ceoncetration, MDIZ: mean diameter of inhibition zone, EC50: effective concentration, IC50: inhibiton concentration.

**Table 2 ijms-20-02926-t002:** Molecular mechanisms underlying bioactivities of costunolide.

Type	Experimental Model	Dose/Concentration	Mechanism of Action	Ref.
Antioxidant effect	STZ-induced diabetic rats	20 mg/kg day	Decreased in TBARS level; increased in GSH content	[8]
	MCF-7, MDA-MB-231	20, 40 μM	Decreased in TBARS level; increased in SOD, catalase, GPx activity	[10]
	5-FU-induced IM	5, 20 mg/kg	Increased in SOD level	[11]
	H_2_O_2_-stimulated PC12 cells	50, 100 μM	Decreased intracellular ROS	[12]
Anti-inflammatory effect	Cg-induced edema; LPS-induced fever	0.015, 0.15, 0.3 mg/kg	Inhibited edema formation; Reduced the fever index	[13]
	LPS-stimulated RAW264.7 cells	0.5, 1.5, 3 μg/ml	Inhibited NF-κB activity, phosphorylation of IκBα and NO production; suppressed iNOS mRNA expression	[14]
	5-FU-induced IM	5, 20 mg/kg	Decreased the expression of iNOS, COX-2, TNF-α and NO	[11]
	IL-22 or IFN-γ-stimulated keratinocytes	12.5 μM	Inhibited STAT1/3 phosphorylation	[15]
	IL-6-stimulated THP-1 cells	6, 12, 25 ng/ml	Inhibited STAT3 and JAK1/2 phosphorylation	[16]
	Ethanol-induced gastric ulcer	5, 20 mg/kg	Suppressed the activation of NF-κB, TNF-α, COX-2, NO and iNOS	[17]
	LPS-stimulated RAW264.7 cells	0.1, 0.3, 1, 3 μM	Suppressed the protein and mRNA expression of IL-1β; inhibited the activity of AP-1 and the phosphorylation of MAPKs	[18]
	Carrageenan-induced pleurisy	5, 10, 15 mg/kg	Reduced accumulation of PMNs and expression of T TNF-α, ICAM-1, P-selectin and nitrotyrosine	[19]
	LPS-stimulated RAW264.7 cells	0.1, 0.5, 1 μM	Induced HO-1 expression and Nrf2 nuclear accumulation; inhibited production of TNF-α and IL-6	[21]
	CD3/CD28-stimulated CD4^+^ T cells	0.5, 1, 2 μM	Inhibited the expression of T-bet, GATA3 and RORγt; suppressed the proliferation of CD4^+^ T cells and expression of CD69; decreased the phosphorylation of ERK and p38	[23]
Antiallergic effect	TNF-α/IFN-γ-stimulated HaCaT cells	2.5, 5, 10 μM	Inhibited the expression of TARC, MDC, RANTES and IL-8	[25]
	IgE-sensitized RBL-2H3	10 μM	Inhibited the expression of β-hexosaminidase	[26]
	OVA-induced mouse asthma model	10 mg/kg	Reduced eosinophil filtration, inflammation score and mucin secretion; decreased the expression of IL-4 and IL-13	
	Ketotifen-stimulated RBL-2H3	0.32, 1.6, 8, 40 μM	Inhibited the release of β-hexosaminidase	[27]
	IL-5-stimulated Y16 cells	0.16, 0.8, 4, 20, 40 μM	Inhibited the proliferation Y16 cells	
Bone remodeling	MC3T3-E1 cells differentiation	10 μM	Increased ALP activity, collagen deposition and mineralization	[29]
	C3H10T1/2 cells differentiation	1, 10, 10^2^, 10^3^, 10^4^ ng/ml	Increased the expression of Dlx5, Runx2, ALP, and OC; reduced the activity of ATF4 and expression of HO-1	[30]
	RANKL-induced osteoclast differentiation	5 μM	Suppressed NFATc1 expression and c-Fos activity	[31]
Neuroprotective agent	DA-stimulated SH-SY5Y	0.8, 4, 2 μM	Decreased the expression of ASYN; increased the expression of Nurr1, VMAT2 and DAT	[33]
	LPS-stimulated BV2 microglial cells	1 μM	Attenuated the expression of TNF-α, IL-1,6, iNOS, MCP-1 and COX-2; inhibited the activation of NF-κB	[34]
Treatment of alopecia	Testosterone-stimulated hHFDPCs	3 μM	Promotes the growth of hHFDPCs; inhibits the 5α-reductase activity	[46]
	Hair growth in mice	3 μM/L	Improved the hair growth	
Inhibition of proliferation	MCF-7 breast cancer cells	10, 100 nM	Inhibited the cell growth; stimulated tubulin assembly	[47]
	K562 leukemia cells	15 μM	Induced cell cycle arrest; induced apoptosis	[48]
	S480 colon cancer cells	5 μM	Suppressed cyclin D1, survivin, β-catenin, and galectin-3; inhibited proliferation and survival of cells	[49]
	LNCaP, PC-3, DU-145 prostate cancer cells	1.3 μM	Inhibited cell proliferation; induced cell cycle arrest at the G1phase	[50]
	HA22T/VGH hepatocellular carcinoma cells	5 μM	Caused G2/M arrest; up-regulated phosphorylation of Chk2, Cdc25c, Cdk1, and cyclin B1	[51]
	HCT-116 colorectal cancer cells	10, 20, 40 μM	Inhibited proliferation; suppressed mTOR phosphorylation and GLS1 activity	[52]
	SK-MES-1 lung squamous carcinoma cells	40, 80 μM	Inhibited growth of cells; induced cell cycle arrest at G1/S phase; upregulated expression of p53 and Bax; downregulated Bcl-2 expression; activated caspase-3	[53]
	SGC-7901 gastric adenocarcinoma cells	20, 40 μM	Arrested cell cycle at G2/M phase; activated caspase-3	[54]
	MCF-7, MDA-MB-231 breast cancer cells	0.9, 1.3, 2.2 μg/mL	Arrested cell cycle at G2/M phase; induced p53 and 14-3-3 expression; inhibited c-Myc, p-Akt and p-BID expression	[55]
	MDA-MB-231 breast cancer cells	15 μM	Induced G2/M cell cycle arrest; upregulated p21WAF1 expression; inhibited cdc2 and cyclin B1 expression	[56]
	MCF-7, MDA-MB-231 breast cancer cells	40 μM	Arrested cell cycle arrest at G2/M phase; inhibited the expression of cyclin D1, D3, CDK-4, CDK-6, p18 INK4c, p21 CIP1/Waf-1 and p27 KIP1	[57]
	K562/ADR chronic myeloid leukemia cells	0.1, 1, 10, 100 μM	Sensitized K562 cells to doxorubicin; inhibited PI3K/Akt activity	[58]
	Eca-109 human esophageal cancer cells	40, 80 μM	Induced cell cycle arrest in G1/S phase; upregulated the expression of p53, p21, Bax and caspase-3; downregulated Bcl-2	[59]
Mitochondria-mediated apoptosis	PC-3, DU-145 prostate cancer cells	20 μM	Enhanced doxorubicin to change of MMP; increased Bax expression and cytochrome c release	[60]
	T24 human bladder cancer cells	25, 50 μM	Increased expression of Bax, downregulated Bcl-2 and surviving; activated caspase-3 and PARP	[48]
	U937 human promonocytic leukemia cells	5, 10	Increased the activation of JNK; inhibited the expression of Bcl-2; induced DNA fragmentation	[61]
	SKOV3, A2780, MPSC1 ovarian cancer cells	10, 20, 30 μM	Triggered the activation of caspase-3, -8, and -9; down-regulated Bcl-2 expression,	[62]
	11Z human epithelial endometriotic cells	IC_50_ 14.21 μM	Induced the activation of caspase-3, -8, and -9; inhibited the activation of Akt and NF-κB	[63]
	ovarian cancer cell line, OAW42-A	12.5, 25, 50 μM	Reduced the mitochondrial membrane potential; increased protein expression of LC3 II and beclin 1	[64]
ER stress-mediated apoptosis	A549 lung adenocarcinoma cells	10, 20, 30 μM	Activated UPR signaling pathways; upregulated GRP78 and IRE1α expression; induced ASK1 and JNK activation	[65]
	U2OS human osteosarcoma cells, A549 human alveolar adenocarcinoma cells, Hela cells	10, 20, 30 μM	Increased expressions of Bip and IREa; increased expressions of p-ASK1, p-JNK and p-ERK; induced generation of Ca^2+^	[66]
	HCT-116, HT-29, SW620 colon cancer cells	10, 20, 30 μM	Inhibited the activity of TrxR1; induced the expression of p-eIF2a, ATF4 and CHOP	[67]
Death receptor mediated apoptosis	NALM-6 human B cell leukemia cell	10 μM	Increased the phosphorylation of FADD; activated caspase-8	[68]
TERT inhibition	NALM-6 human B cell leukemia cell	10 μM	Suppressed telomerase activity; inhibited the expression of hTERT mRNA and protein	
	MCF-7, MDA-MB-231 breast cancer cells	10, 50, 80, 100 μM	Inhibited the cell growth, telomerase activity and hTERT mRNA expression; inhibited bindings of hTERT promoters; inhibited the expression of c-Myc and Sp1	[69]
	A172, U87MG, T98G glioma cells	10, 20, 30, 40 μM	Decreases Nrf2 levels; Suppressed telomerase activity; decreased expression of G6PD and TKT	[70]
	A172, U87MG glioma cells	30 μM	Inhibited hTERT expression	[71]
	HepG2/C3A, PLC/PRF/5 HCC cells	5, 10, 50 μM	Inhibited AFP secretion and mRNA expression; decreased cell migration	[72]
Inhibition of angiogenesis	subcutaneous murine sponge model	5, 10, 20 mg/kg	Reduced hemoglobin concentration and VEGF levels	[74]
	VEGF-stimulated HUVECs	IC_50_ 5.7 μM	Inhibited VEGF-induced proliferation and migration; inhibited the VEGF-induced autophosphorylation of KDR/Flk-1	[75]
	AGS, Caco-2, HepG2/C3A cancer cells	10 μM	Decreased VEGF secretion and mRNA levels	[76]
Inhibition of tumor metastasis	MDA-MB-231 breast cancer cells	20 μM	Inhibited TNF𝛼-induced cells migration and invasion; reduced phosphorylation of IKK and I𝜅B𝛼; inhibited p65 NF-𝜅B subunit	[77]
	IMR-32, LA-N-1, SK-N-SH neuroblastoma cell	0.1, 1, 10 μM	Inhibited migration and invasion; suppressed MMP2 expression	[78]
	SW-872, SW-982, TE-671 soft tissue sarcomas	3, 10, 20 μg/mL	Inhibited the invasion potential; changed the expression of MMPs	[79]
	TR-LE (temperature-sensitive rat lymphatic endothelial) cells	IC_50_ 1.37 μM	Suppressed cell proliferation; inhibited capillary-like tube formation	[80]
	MDA-MB-157, MDA-MB-436, Bt-549 breast cancer cells	10, 25 μM	Reduced detyrosinated tubulin; decreased microtentacle (McTN) frequency; reduced tumor cell attachment	[81]
Antidiabetic effect	α-Amylase, α-Glucosidase, fructosamine formation, glycation	IC_50_ 5.88 or 67.5 μM	Inhibited the activity of α-Amylase, α-Glucosidase; inhibited fructosamine formation;	[82]
	streptozotocin-induced diabetic rats	5, 10, 20 mg/kg	Reduced glucose levels and HbA_1c_; increased insulin levels; reduced cholesterol, TG, LDL; increased HDL	[8]

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
