# Peer review of "Costunolide—A Bioactive Sesquiterpene Lactone with Diverse Therapeutic Potential"

_ijms, 2019, doi:10.3390/ijms20122926_

Round 1

Reviewer 1 Report

The authors present a review of literature on Costunolide, a sesquiterpene lactone with pharmacological potential. The work presented need some major in order to be published.

The references should be more recent. Even if there are some new articles cited, the average number of articles are quite old, and this could and should be improved. In some cases the references are clearly missing. See for example, “Studies have shown that costunolide improves glycemic control and lipid battery in experimental models of diabetes”. What studies? Are there more, or just one?

In most parts, the presented article seems to be constructed only on the article’s abstracts. It looks like a summary of works added together without a proper critical analysis and an added value based on the authors experience and critical thinking. In this view, the review need a lot of work and major modifications to be a work of quality.

The authors presented a sum of effects of Costunolide in vitro or in vivo, but they did not provide the doses administered or the output of each assay. It is very hard to really understand the pharmacological potential of this substance without proper quantitative assays. For example: “The compound significantly reduced blood glucose level, glycosylated hemoglobin (HbA1c), serum total cholesterol, triglyceride and LDL cholesterol, and remarkably increased plasma insulin, tissue glycogen, HDL cholesterol and serum protein in streptozotocin (STZ)-induced diabetic rats”. What dose produced this effect? It reduced blood glucose level by how much? What substance was used as control? Is a significant difference?

Some toxicological data should be added. As promising as it is, surely this compound was tested and assayed for its toxic effects. A short section and discussion on this is strongly needed.

The authors should clearly state, when a crude extract was tested or a purified costunolide. Please check all the manuscript and all the source materials and provide the proper information if a mixture or extract was used in the biological assay.

The review need also the authors input from their own experience on this field. The reader should this.

There are many typing mistakes (5-floirouracil ??) and the manuscript was not really finished. See rows 102, 114, 231! “Cite a reference”???. The manuscript should be properly prepared for the journals requirements.

Author Response

Author's Reply to the Review Report (Reviewer 1)

Reviewer #1: The references should be more recent. Even if there are some new articles cited, the average number of articles are quite old, and this could and should be improved. In some cases the references are clearly missing.

Author reply: We would like to thank this reviewer for his thorough and critical reviewing of the manuscript. During submission, we inadvertently uploaded the premature version of the manuscript, where many citations were missing. We have paid careful attention to update all citations and also incorporated few recent articles on anticancer effects of costunolide in the revised manuscript.

Reviewer #1: See for example, “Studies have shown that costunolide improves glycemic control and lipid battery in experimental models of diabetes”. What studies? Are there more, or just one?

Author reply: The effect of costunolide on the glycemic control was studied by only one research group. The study was on streptozotocin-induced rat diabetes model. We have mentioned that additional studies are warranted to confirm this result.

Reviewer #1: In most parts, the presented article seems to be constructed only on the article’s abstracts. It looks like a summary of works added together without a proper critical analysis and an added value based on the authors experience and critical thinking. In this view, the review need a lot of work and major modifications to be a work of quality. The authors presented a sum of effects of Costunolide in vitro or in vivo, but they did not provide the doses administered or the output of each assay. It is very hard to really understand the pharmacological potential of this substance without proper quantitative assays. For example: “The compound significantly reduced blood glucose level, glycosylated hemoglobin (HbA1c), serum total cholesterol, triglyceride and LDL cholesterol, and remarkably increased plasma insulin, tissue glycogen, HDL cholesterol and serum protein in streptozotocin (STZ)-induced diabetic rats”. What dose produced this effect? It reduced blood glucose level by how much? What substance was used as control? Is a significant difference?

Author reply: We appreciate reviewer’s opinion in this regard. Because the manuscript is quite comprehensive, it would be more lengthy if every detail of each study is incorporated. We have surely addressed each study outcomes and co-related and/or contrasting findings by others. As a whole, we opted to provide overall pharmacological and therapeutical value of costunolide in an integrated way. 

Reviewer #1: Some toxicological data should be added. As promising as it is, surely this compound was tested and assayed for its toxic effects. A short section and discussion on this is strongly needed.

Author reply: There have no published acute and chronic toxicity studies with costunolide. However, a recent article reporting the genotoxic potential of costunolide has been included in the revised version.

Reviewer #1: The authors should clearly state, when a crude extract was tested or a purified costunolide. Please check all the manuscript and all the source materials and provide the proper information if a mixture or extract was used in the biological assay.

Author reply: We have checked each reference to make sure the experimental results of costunolide have been reported, any effect of extracts have been mentioned under extracts effect.

Reviewer #1: The review need also the authors input from their own experience on this field. The reader should this.

Author reply: We have cited our published works on hair growth promoting effect of costunolide.

Reviewer #1: There are many typing mistakes (5-floirouracil ??) and the manuscript was not really finished. See rows 102, 114, 231! “Cite a reference”???. The manuscript should be properly prepared for the journals requirements.

Author reply: We have carefully corrected typographical and grammatical errors.

Reviewer 2 Report

The review entitled “Costunolide – A bioactive sesquiterpene lactone with diverse therapeutic potential” is aimed to summarize the recent researches on costunolide, a sesquiterpene lactone, focusing on biological activities and their underlying mechanisms.

Minor revisions should be made, and the manuscript should be completed and/or modified as follows:
1. The authors are advised to remove and from keywords
2. The authors should rephrase the following: „A wide variety of non-nutritive plant constituents have been proven to be effective therapy for many disease processes including, but not limited to, infections, diabetes, heart diseases, neurological disorders and cancer” (lines 23-25).
3. The authors should rephrase the following: „One of the major classes of bioactive phytochemicals is the terpenoids, which are classified according to their carbon numbers as monoterpenes, diterpenes, sesquiterpenes and triterpenes.” (lines 28-30). The carotenoids are terpenoids, too.
4. The authors should carefully check the agreement of subject and verb, for example: „It is known that the largest group of sesquiterpene lactones is germacranolides” (lines 34-35), „Germacranolides is a 10,5-ring structure” (line 35), line 415, etc.
5. The reference numbers should be placed before the punctuation (line 41, 238, 240, 308, 329, etc)
6. The authors should rephrase the following: „Structurally, costunolide is a monocarboxlic acid having three double bonds which by catalytic hydrogenation generates hexahydrocostunolide (C15H26O2) (Figure 1)”. (lines 41-42). One can understand that Fig. 1 presents the chemical structure of hexahydrocostunolide.
7. The authors should carefully check the references 8 and 9 from text (line 57 and 59). Reference 9 presents the effects of costunolide on GSH level.
8. The authors should correct „5-floirouracil” (line 65), „alzheimer's” (line 69), „redcued” (line 159), „antiinflammatory” (line 179), „Anti-microbial” (line 182, 207), „costunilode” (line 403)
9. The authors should rephrase the following: „Treatment of lipopolysaccharide (LPS)-treated Raw264.7 cells with costunolide resulted in diminished activation” (lines 73-75)
10. The authors should rephrase the following: „Treatment of human THP-1 cells with costunolide inhibited interleukin (IL)-6-induced phosphorylation and the DNA binding activity of another redox regulated transcription factor, signal transducer and activator of transcription (STAT)- 3 via h downregulation of Janus-activated kinase (JAK)-1 and-2, and tyrosine kinase-2 (Tyk2) phosphorylation” (lines 81-85)
11. The authors should carefully check the line 95, „Prawan et al 2005” is not cited as reference. The same with line 131 „Khosla, 2009, 818-820”
12. The authors should carefully check the lines 102, 114, 231: „(Cite a reference)”
13. The authors should rephrase the following: „the elimination of reactive oxygen species (ROS) can be a potential strategy to delay the disease progression” (lines 172-174). The elimination of ROS is not possible, maybe the reduction.
14. The authors should use the italic style (in vivo, in vitro): line 192, 215, 225, 402.
15. Line 369: the authors should check (91)

16. The authors should rephrase the following: „A number of PTP1B inhibitors, synthesized or isolated as bioactive natural products, have been to stimulate insulin signaling” (lines 372-374)
17. The authors should carefully check the references and describe them according to instructions for authors (reference 1).

Author Response

Author's Reply to the Review Report (Reviewer 2)

Thank you very much for thorough and careful reviewing of the manuscript. We appreciate your valuable comments.

1. The authors are advised to remove and from keywords

Author reply: Done.

2. The authors should rephrase the following: „A wide variety of non-nutritive plant constituents have been proven to be effective therapy for many disease processes including, but not limited to, infections, diabetes, heart diseases, neurological disorders and cancer” (lines 23-25).

Author reply: We have rephrased the sentence as “A wide variety of non-nutritive plant constituents, commonly known as phytochemicals, are being used as therapy for many disease processes including, but not limited to, infections, diabetes, heart diseases, neurological disorders and cancer.”

3. The authors should rephrase the following: „One of the major classes of bioactive phytochemicals is the terpenoids, which are classified according to their carbon numbers as monoterpenes, diterpenes, sesquiterpenes and triterpenes.” (lines 28-30). The carotenoids are terpenoids, too.

Author reply: We have rephrased the sentence as “One of the major classes of bioactive phytochemicals is the terpenoids, which are widely present in various plants and marine organisms, and are being examined for developing new antifungal, anticancer, anti-inflammatory and antiviral agents [2]..”

4. The authors should carefully check the agreement of subject and verb, for example: „It is known that the largest group of sesquiterpene lactones is germacranolides” (lines 34-35), „Germacranolides is a 10,5-ring structure” (line 35), line 415, etc.

Author reply: To comply with subject-verb agreement, we have rephrased the sentence as “The largest group of sesquiterpene lactones is germacranolides [3], which possesses a 10,5-ring structure and is present in several plant families”

5. The reference numbers should be placed before the punctuation (line 41, 238, 240, 308, 329, etc)

Author reply: This has been corrected.

6. The authors should rephrase the following: „Structurally, costunolide is a monocarboxlic acid having three double bonds which by catalytic hydrogenation generates hexahydrocostunolide (C15H26O2) (Figure 1)”. (lines 41-42). One can understand that Fig. 1 presents the chemical structure of hexahydrocostunolide.

Author reply: This has been changed as “Structurally, costunolide (Figure 1) is a monocarboxlic acid having three double bonds which by catalytic hydrogenation generates hexahydrocostunolide.

7. The authors should carefully check the references 8 and 9 from text (line 57 and 59). Reference 9 presents the effects of costunolide on GSH level.

Author reply: Thank you for pointing out this error. The switching of reference is corrected.

8. The authors should correct „5-floirouracil” (line 65), „alzheimer's” (line 69), „redcued” (line 159), „antiinflammatory” (line 179), „Anti-microbial” (line 182, 207), „costunilode” (line 403)

Author reply: These and other typographical errors have been corrected.

9. The authors should rephrase the following: „Treatment of lipopolysaccharide (LPS)-treated Raw264.7 cells with costunolide resulted in diminished activation” (lines 73-75)

Author reply: The sentence has been rephrased as “Costunolide negated NF-kB activation via blockade of IkBa phosphorylation in lipopolysaccharide (LPS)-stimulated Raw264.7 cells, thereby reducing the expression of proinflammatory markers, such as inducible nitric oxide synthase (iNOS), tumor necrosis factor-alpha (TNF-a), and the production of nitric oxide (NO) [14].”

10. The authors should rephrase the following: „Treatment of human THP-1 cells with costunolide inhibited interleukin (IL)-6-induced phosphorylation and the DNA binding activity of another redox regulated transcription factor, signal transducer and activator of transcription (STAT)- 3 via h downregulation of Janus-activated kinase (JAK)-1 and-2, and tyrosine kinase-2 (Tyk2) phosphorylation” (lines 81-85)

Author reply: The sentence has been rephrased as, “Likewise, treatment of human THP-1 cells with costunolide inhibited interleukin (IL)-6-induced phosphorylation and the DNA binding activity of signal transducer and activator of transcription (STAT)-3 via downregulation of Janus-activated kinase (JAK)-1 and-2, and tyrosine kinase-2 (Tyk2) phosphorylation [16].

11. The authors should carefully check the line 95, „Prawan et al 2005” is not cited as reference. The same with line 131 „Khosla, 2009, 818-820”

Author reply: These citations have been updated.

12. The authors should carefully check the lines 102, 114, 231: „(Cite a reference)”

Author reply: These citations have been updated.

13. The authors should rephrase the following: „the elimination of reactive oxygen species (ROS) can be a potential strategy to delay the disease progression” (lines 172-174). The elimination of ROS is not possible, maybe the reduction.

Author reply: We have replaced the word “elimination’ with “reduction”

14. The authors should use the italic style (in vivo, in vitro): line 192, 215, 225, 402.

Author reply: Done as per journal style.

15. Line 369: the authors should check (91)

Author reply: This has been deleted.

16. The authors should rephrase the following: „A number of PTP1B inhibitors, synthesized or isolated as bioactive natural products, have been to stimulate insulin signaling” (lines 372-374)

Author reply: This has been deleted.

17. The authors should carefully check the references and describe them according to instructions for authors (reference 1).

Author reply: Thank you very much. We have carefully checked the references for appropriateness.

Round 2

Reviewer 1 Report

The authors modified just a small portion of the problems mentioned in the first review and even if they declared they did modified and corrected, they did not. I still maintain my comments that the manuscript looks like a sum of abstracts added together without a proper logical flow.

I think the authors should add, when possible, the tested doses for the extracts. It is very important to understand the real and true pharmacological potential. They declare that "it would be more lengthy if every detail of each study is incorporated". Is not true! It would be quite easy and simple! Take for example table 1. Just add a column with the active dose, like MIC. The active doses should be added in the whole manuscript and the authors should do the effort to read all the references cited, and not just the abstracts.

Author Response

Author's Reply to the Review Report (Reviewer 1)

Thank you very much for thorough and careful reviewing of the manuscript. We appreciate your valuable comments.

Reviewer #1:
The authors modified just a small portion of the problems mentioned in the first review and even if they declared they did modified and corrected, they did not. I still maintain my comments that the manuscript looks like a sum of abstracts added together without a proper logical flow.

I think the authors should add, when possible, the tested doses for the extracts. It is very important to understand the real and true pharmacological potential. They declare that "it would be more lengthy if every detail of each study is incorporated". Is not true! It would be quite easy and simple! Take for example table 1. Just add a column with the active dose, like MIC. The active doses should be added in the whole manuscript and the authors should do the effort to read all the references cited, and not just the abstracts.

Author reply: Thank you for your comments. We reviewed each article in its entirety. As in your comment, we added an effect concentration to Table 1. And table 2( page 20~24 ) is newly created. The modifications are shown in red text. We have also commissioned English proofing at https://www.editage.co.kr. I would appreciate your comments if there is still a shortage.

Reviewer 2 Report

The authors made all the required changes and the manuscript has been significantly improved.

Author Response

Author's Reply to the Review Report (Reviewer 2)

Thanks a lot for your delicate comment

Reviewer #2: The authors made all the required changes and the manuscript has been significantly improved.

Author reply:  Thank you for your good comment.

Round 3

Reviewer 1 Report

the manuscript can be published.